# Identification of Potential Probiotics Producing Bacteriocins Active against *Listeria monocytogenes* by a Combination of Screening Tools

**DOI:** 10.3390/ijms22168615

**Published:** 2021-08-10

**Authors:** Christian K. Desiderato, Steffen Sachsenmaier, Kirill V. Ovchinnikov, Jonas Stohr, Susanne Jacksch, Dominique N. Desef, Peter Crauwels, Markus Egert, Dzung B. Diep, Oliver Goldbeck, Christian U. Riedel

**Affiliations:** 1Institute of Microbiology and Biotechnology, University of Ulm, Albert-Einstein-Allee 11, 89081 Ulm, Germany; christian.desiderato@uni-ulm.de (C.K.D.); federer.sachsenmaier@gmail.com (S.S.); jonas.stohr@web.de (J.S.); dominique.desef@uni-ulm.de (D.N.D.); petcrauwels@hotmail.com (P.C.); oliver.goldbeck@uni-ulm.de (O.G.); 2Faculty of Chemistry, Biotechnology and Food Science, Norwegian University of Life Sciences, Universitetstunet 3, 1433 Ås, Norway; kirill.ovchinnikov@nmbu.no (K.V.O.); dzung.diep@nmbu.no (D.B.D.); 3Faculty of Medical and Life Sciences, Institute of Precision Medicine, Furtwangen University, Campus Schwenningen, Jakob-Kienzle-Straße 17, 78054 Villingen-Schwenningen, Germany; Susanne.Jacksch@hs-furtwangen.de (S.J.); Markus.Egert@hs-furtwangen.de (M.E.)

**Keywords:** *Listeria monocytogenes*, food, control, antimicrobial peptide, bacteriocin, lactic acid bacteria

## Abstract

*Listeria monocytogenes* is an important food-borne pathogen and a serious concern to food industries. Bacteriocins are antimicrobial peptides produced naturally by a wide range of bacteria mostly belonging to the group of lactic acid bacteria (LAB), which also comprises many strains used as starter cultures or probiotic supplements. Consequently, multifunctional strains that produce bacteriocins are an attractive approach to combine a green-label approach for food preservation with an important probiotic trait. Here, a collection of bacterial isolates from raw cow’s milk was typed by 16S rRNA gene sequencing and MALDI-Biotyping and supernatants were screened for the production of antimicrobial compounds. Screening was performed with live *Listeria monocytogenes* biosensors using a growth-dependent assay and pHluorin, a pH-dependent protein reporting membrane damage. Purification by cation exchange chromatography and further investigation of the active compounds in supernatants of two isolates belonging to the species *Pediococcus acidilactici* and *Lactococcus garvieae* suggest that their antimicrobial activity is related to heat-stable proteins/peptides that presumably belong to the class IIa bacteriocins. In conclusion, we present a pipeline of methods for high-throughput screening of strain libraries for potential starter cultures and probiotics producing antimicrobial compounds and their identification and analysis.

## 1. Introduction

*Listeria monocytogenes* is a saprophytic soil bacterium and an important intracellular pathogen of humans and animals [1]. Due to its versatile lifestyle and ability to grow under a wide range of stressful conditions, *L. monocytogenes* is highly competitive in the environment, under conditions of food production and preservation, in different food matrices and the gastrointestinal tract of the host [2,3]. Listeriosis (i.e., the disease caused by *L. monocytogenes*) is usually acquired upon consumption of contaminated food products. In otherwise healthy individuals, the course of the disease is relatively mild or even asymptomatic. However, in high-risk groups (e.g., pregnant women, newborns, elderly people, immunosuppressed patients, etc.), the disease may be very severe and even fatal [4]. A clear and steady increase in the number of cases of listeriosis has been observed in recent years [5]. In 2014, there were 2206 confirmed cases of listeriosis in humans in the European Union, with a mortality rate of approximately 18% [5]. This makes listeriosis by far the deadliest food-borne disease in humans.

One of the reasons for the rising numbers of listeriosis outbreaks is an increased demand of consumers for ready-to-eat and minimally processed foods, which are particularly vulnerable to contaminations with foodborne pathogens. At the same time, reduced shelf life of such products leads to an increase in production costs, prices for consumers, and the amount of food waste. Therefore, the development of new methods for cost-effective and efficient preservation that, at the same time, maintain taste, texture, sensory, and nutritional properties, are of considerable interest. A green-label approach for food preservation that meets most of these criteria are antimicrobial peptides (AMPs) termed bacteriocins [6,7,8]. Additionally, bacteriocin-producing bacteria have been shown to protect against infections with pathogenic microorganisms [9,10,11,12,13] and is thus considered an important criterion for the selection of probiotic bacteria [14,15].

Bacteriocins are ribosomally synthesized antimicrobial peptides naturally produced by a wide range of bacteria, especially of lactic acid bacteria (LAB), and released into the extracellular environment [16,17]. They are able to suppress growth or directly kill target organisms in a specific manner and their biological role is to provide a competitive advantage of the producing organisms in an ecological niche [17]. Based on size, presence of (extensive) posttranslational modifications, and thermostability bacteriocins are categorized into three main classes with various subclasses [8,18]. Class I and II bacteriocins are small (<10 kDa), heat-stable peptides that differ in the extent of their post-translational modification. Class I bacteriocins are modified by dedicated enzymatic modification machineries resulting in unusual amino acid residues. Similar modifications are absent in class II bacteriocins. They may, however, be translated as (inactive) pre-peptides, activated upon transport by cleavage of the leader peptides and contain stabilizing disulfide bonds. In contrast, class III contains large, heat-labile antimicrobial peptides.

Most bacteriocins are active against organisms closely related to the producer, which are immune against their own bacteriocin. Nevertheless, some bacteriocins have a broader spectrum of target organisms [16]. Depending on the mechanism, bacteriocins are either bacteriostatic and/or bactericidal for their targets. For nisin and other class I bacteriocins, it has been shown that their activity is mediated via lipid II [19,20], which is an essential carrier molecule for cell wall precursors. The bacteriostatic activity is achieved by the binding of nisin to lipid II. This inhibits cell wall synthesis and thus growth of target organisms, even at lower concentrations. At higher concentrations, lipid II and nisin form pore complexes in the cytoplasmic membrane, leading to membrane depolarization and killing of the target. Pore formation has also been described as the mechanism of action of class II bacteriocins. There are several receptors described for different class II bacteriocins including mannose phosphotransferase systems (PTS^Man^) [21,22,23,24] and maltose ATP-binding cassette transporter [25]. However, the exact mechanisms or pore formation of these class II bacteriocins is largely unknown. It is currently under debate if the pores are formed exclusively by the peptides after receptor binding or if pores are formed in complex with other molecules.

For food preservation, bacteriocins can either be produced ex situ and added during or after production. Alternatively, bacteriocin-producing bacteria may be added during or after production as protective cultures or probiotic supplements [26]. Despite their advantageous properties, merely a handful of bacteriocins are currently approved and marketed for use in food [17,26]. This highlights the need for rapid, reliable, and cost-efficient methods for the identification and characterization of novel bacteriocins and their producers. Ideally, the screening methods already provide information on the potential mechanisms. In most studies, bacteriocin-producing bacteria are identified by classical plating methods and measurements of inhibition zones of an indicator organism around a colony of the producer [27]. Alternatively, there are a number of bioinformatic tools for in silico prediction of bacteriocin gene clusters in genomic and metagenomic datasets [28,29,30].

We recently generated *L. monocytogenes* biosensors expressing the ratiometric pH-sensitive fluorescent protein pHluorin that allow for easy monitoring of membrane-damaging activity of bacteriocins [31]. The pHluorin protein has a bimodal excitation spectrum [32] and relative fluorescence intensity at the two excitation peaks shifts in response to changes in pH in a ratiometric manner [32]. Under steady-state conditions, *L. monocytogenes* possesses an intracellular pH (pH_i_) of 7.6–8.0 [33]. However, when placed in a buffer with acidic pH and exposed to compounds that disrupt membrane integrity, the internal pH_i_ will rapidly change to the extracellular pH of the buffer, and this shift can be detected by the changes in the relative intensity of fluorescence after excitation at 400 and 470 nm. Thus, in an acidic buffer system, the *L. monocytogenes* biosensor can be used to detect compounds with membrane damaging activity [31].

Here, these biosensors were used to screen supernatants of a library of bacterial isolates from raw cow’s milk for growth inhibition and membrane damaging activity. This allowed for the identification of probiotic candidates that produce antimicrobial compounds. The antimicrobial compounds were further purified and characterized.

## 2. Results

### 2.1. Isolation and Typing of LAB from Raw Cow Milk

The collection of 55 raw milk isolates was typed by 16S rRNA gene sequencing and MALDI-Biotyping (Appendix A). According to 16S rRNA gene sequences, all isolates were identified to the species level and belong to genera of the lactic acid bacteria except for isolates 14 and 16, which were typed as *Corynebacterium glutamicum*. The results obtained by MALDI-Biotyping were mostly in agreement with the 16S rRNA gene sequences, but generally less accurate. For 13 isolates, no identification was achieved by MALDI-Biotyping as no matching spectrum was found in the database. For one isolate, identification was only possible on the genus level and for isolates 7, 10, 22, 24, 34, 35, 39, 50, and 56, the results of the MALDI-Biotyping contradicted identification based on 16S rRNA gene sequences. The positive control, *P. acidilactici* 347, was typed as *P. lolii* or *Pediococcus* spp. by 16S rRNA gene sequencing and MALDI typing, respectively. According to the List of Prokaryotic Names with Standing in Nomenclature of the German Collection of Microorganisms and Cell Cultures, *P. lolii* is synonymous with *P. acidilactici* [34]. Thus, all isolates identified as *P. lolii* were termed *P. acidilactici* in this study. Isolate 34 was typed as *L. garvieae*, according to the MALDI-TOF results as its 16S rRNA gene sequence had the same identity to the NCBI database sequences for *L. petauri* and *L. garvieae*. In total, bacteria of six different genera and 10 different species were identified and the most prevalent species isolated were *L. lactis* (14 isolates) and *L. garvieae* (13 isolates).

### 2.2. Screening of Supernatants for Antimicrobial Activity

All isolates were grown on MRS medium, and supernatants were collected and screened for antimicrobial activity using *L. monocytogenes* EGD-e/pNZ-P_help_-pHluorin [31] as a sensor in two different assays (Figure 1). Supernatants of 27 isolates (i.e., 49%) were able to inhibit growth of the sensor strain significantly compared to supernatants of *L. lactis* MG1363 (i.e., a strain that does not produce a bacteriocin). However, the levels of inhibition varied greatly between isolates. Complete inhibition of growth of the sensor strain was only observed for six isolates (11%) and the pediocin-producing control strain *P. acidilactici* 347.

Similar results were obtained using the ratiometric pH-dependent fluorescence of pHluorin as a read out for membrane damage. Here, the number of supernatants that produced an effect on membrane integrity was markedly lower. Of the 55 tested supernatants, only nine (i.e., 16%) significantly reduced the fluorescence ratio of pHluorin compared to the supernatant of the non-producer *L. lactis* MG1363 (Figure 1B). Interestingly, those were also the top nine strains that showed the most efficient growth inhibition. These top nine isolates belong to the species *P. acidilactici* (isolates M1, M3, 13, and 27), *E. faecium* (isolates M4 and M9), *L. lactis* (isolates M7 and 18), and *L. garvieae* (isolate 17) all of which contain natural producers of a variety of bacteriocins [18].

### 2.3. Characterization of the Antimicrobial Activity of Representative Isolates

To confirm the results of the screening, antimicrobial activity was estimated by growth inhibition assays with serial dilutions of the supernatants of o/N cultures of two representative isolates of the species *P. acidilactici* and *L. garvieae*. Supernatants of *P. acidilactici* M1 and *L. garvieae* 17 contained high levels of antimicrobial activity comparable or only marginally lower than those observed for *P. acidilactici* 347 (Figure 2). Further experiments were performed to investigate the kinetics of production of the antimicrobial compound by the two isolates M1 and 17 in comparison to the control strain *P. acidilactici* 347.

All three strains showed distinct profiles of optical density (OD_600_), pH, and antimicrobial activity when cultured in MRS broth (Figure 3). Growth and pH of the two *P. acidilactici* strains M1 and 347 were clearly different with *P. acidilactici* M1 showing considerably slower growth and lower final OD_600_ (Figure 3a). This was also reflected by faster and more pronounced acidification of spent culture medium by *P. acidilactici* 347. *L. garvieae* 17 reached similar final OD_600_ as *P. acidilactici* 347, but showed somewhat slower growth in late exponential growth phase and acidification of the culture broth was not as pronounced.

In line with a more rapid growth, OD-normalized antimicrobial activity increased more rapidly in culture supernatants of *P. acidilactici* 347 compared to *P. acidilactici* M1 (Figure 3b), but both strains reached comparable levels (approx. 750–780 BU ml^−1^ OD^−1^) after 24 h. The kinetics of production of the antimicrobial activity by *L. garvieae* 17 was clearly distinct from the *P. acidilactici* strains. Similar to *P. acidilactici* 347, antimicrobial activity increased during exponential growth until 12 h (2560 BU/mL, data not shown) and remained constant thereafter. However, OD_600_ still increased from approx. 5 to 7 between t = 12 and 24 h, resulting in reduced OD-normalized activity (Figure 3b).

### 2.4. Partial Purification and Characterization of the Antimicrobial Compounds of P. Acidilacici M1 and L. Garvieae 17

For purification and characterization of the antimicrobial compounds produced by *P. acidilactici* M1 and *L. garvieae* 17, supernatant proteins were precipitated and further purified via cation exchange chromatography. Similar to samples of the control strain *P. acidilactici* 347, elution profiles of supernatant proteins of the two isolates contained a prominent peak in absorbance at 214 nm at the onset of elution (Figure 4a). Peak fractions were collected and further analyzed.

Antimicrobial activity was assayed in samples of different steps during the purification process. For all strains, peak fractions of the CIEX chromatography strongly inhibited growth of the sensor strain *L. monocytogenes* EGDe/pNZ-P_help_-pHluorin (Figure 4b). Calculations of BU/mL total activity in each sample and recovery rates relative to the initial activity in untreated supernatants (Table 1) revealed that only about 5% of the total activity was recovered from supernatants of *P. acidilactici* M1 by ammonium sulfate precipitation. Subsequent CIEX chromatography was efficient and about 4.5% of the initial activity was contained in combined fractions 6 and 7. Recovery of the activity was slightly higher (20% after precipitation and 12% in combined CIEX fractions F5 and F6) for *L. garvieae* 17.

Similar to samples of the positive control strain *P. acidilactici* 347, antimicrobial activity of both isolates was efficiently abolished by treatment with proteinase K, but was resistant to heat treatment (Figure 5a). This indicated that the antimicrobial compounds in the supernatants of *P. acidilactici* M1 and *L. garvieae* 17 are heat-stable peptides, potentially bacteriocins.

The species *P. acidilactici* and *L. garvieae* are known to contain strains that produce different class IIa bacteriocins [35,36]. Bacteriocins of this class are known to require PTS^Man^ subunits IIC and IID as receptors to exert their activity. To test if the antimicrobial peptides of isolates M1 and 17 also act in a PTS^Man^-dependent manner, we tested their activity in a spot-on-lawn assay using *L. monocytogenes* EGDe or EGY2 (i.e., an isogenic mutant carrying a 84 bp deletion in the *mptD* gene encoding the IID subunits of the PTS^Man^ that was shown to be resistant to the class IIa bacteriocin mesentericin Y105) [37]. This revealed that wildtype *L. monocytogenes* EGDe is highly sensitive to supernatants as well as CIEX-purified supernatant proteins of both *P. acidilactici* strains and *L. garvieae* 17 (Figure 5b). In contrast, no inhibition was observed with any of the samples when the *mptD* mutant strain EGY2 was used as the sensor.

## 3. Discussion

With the presented study, we aimed at establishing a set of methods for rapid identification of (novel) bacteriocin producers in a strain collection and the subsequent analysis of the compounds and its production. In a first step, a small collection of bacterial strains was isolated and typed to the species level using MALDI-Biotyping and 16S rRNA gene sequence analysis. Of the 55 isolates, all except two belonged to the genera *Lactococcus*, *Enterococcus*, *Leuconostoc*, *Pediococcus*, and *Weissella*. This is in line with those groups of LAB that are highly abundant in raw cow’s milk detected by culture-dependent and independent methods [38] and can be cultivated on the MRS and GM17 agar used for isolation. Additionally, *Corynebacterium* spp. has been detected in cow’s milk, albeit less frequently and in fewer numbers [38].

Typing based on 16S rRNA gene sequences is a more general approach and is considered as a reference method for microorganism identification [39]. Identification using 16S rRNA gene sequences is performed by comparison to comprehensive sequences databases such as NCBI or EzBioCloud [40,41], but requires additional time for PCR amplification and DNA sequencing. In contrast, MALDI-Biotyping is a rapid and cost-effective alternative assay for bacterial identification [42] and is mostly used in the clinical identification of medically important microbial isolates [43,44], but is also promoted for other applications [45,46]. The technique is based on generating mass spectral fingerprints of biomolecules contained in a sample and matching of this spectrum to spectra in a reference database [43,44]. Of the two methods used for typing, 16S rRNA gene sequencing generally produces more robust results, suggesting that the commercially available databases of the MALDI-Biotyping system does not contain enough reference spectra of the bacterial groups isolated to cover the range of spectra within these groups. Similar observations have been made by several studies [42,47,48].

Overall, our screening revealed that supernatants of about 50% of the isolates (total 27) significantly inhibited growth of *L. monocytogenes*. Of these supernatants, nine (i.e., 16%), were also able to elicit membrane-damage. This is in the range of a similar screen that identified 28 bacteriocin producers in a collection of 138 strains isolated from dairy products using M17 and MRS media for isolation and *L. innocua* as an indicator organism in an agar diffusion assay [49].

Results of the growth inhibition assay were more variable. Culture supernatants of lactic acid bacteria have a low pH and contain lactate and other short chain fatty acids. This combination has antimicrobial activity per se [50,51]. This suggests that at least some of the hits identified by the growth inhibition assay may not be bacteriocin producers but rather inhibit the growth of the sensor strain by a combination of SCFA and low pH. Although the results were less variable in pHluorin assays, a number of supernatants also produced a partial or incomplete reduction in the fluorescence ratio of the sensor. This resembles the signals obtained with sublethal concentrations of pediocin and nisin in a previous study [31]. It remains to be investigated whether these intermediate signals are derived from a homogenous population of sensors with partial drop in intracellular pH in all bacteria, or a heterogenous population of sensors in which some bacteria have disrupted membranes and others are completely intact. For further screenings of larger libraries, the pHluorin assay definitely has advantages and yields a faster readout. On the other hand, it does not allow for the identification of bacteriostatic compounds that inhibit growth by mechanisms other than pore formation. Likewise, low concentrations of bacteriocins such as nisin, which have a dual mechanism with growth inhibition at low concentrations and pore formation at higher concentrations, will not be identified using the pHluorin assay. For screenings using growth dependent assays in 96 well format, the pH of the supernatants needs to be considered. This can either be addressed by neutralizing pH prior to the screen, which is laborious for a large number of supernatants, or by performing a secondary screen with neutralized supernatants only for a selection of hits.

Further investigation of the antimicrobial activity of isolates M1 and 17 revealed that the kinetics of production are similar to other bacteriocin producers with a maximum in the late exponential growth phase [52,53,54]. The antimicrobial compounds of isolates M1 and 17 were partially purified using ammonium sulfate precipitation and subsequent cation exchange chromatography of supernatant proteins, although the majority of the activity (approx. 90–95%) was lost mostly during precipitation (Figure 4 and Table 1). A first set of experiments toward characterization of the antimicrobial compounds revealed that they are heat-stable and protease-sensitive, suggesting that they may be bacteriocins. *P. acidilactici* M1 and *L. garvieae* 17 belong to species, which are known to contain strains producing class II bacteriocins such as pediocins and garvicins [21,35,36]. Many class II bacteriocins target the IIC and IID subunits of a defined subgroup of PTS^Man^ [55] and mutational analysis identified a number of amino acid residues in an extracellular domain of the IID subunits of *L. lactis* and *L. garvieae* strains that are critical for susceptibility [21,22]. Although *L. monocytogenes* EGDe encodes four members of the PTS^Man^ family, a mutant (*L. monocytogenes* EGY2) has been described that carries a deletion of 84 bps in the *mptD* gene (*lmo0098*) encoding the IID subunit of PTS^Man^-2 [37]. This mutant is highly resistant to the class II bacteriocin mesentericin Y105. The 84 bp deletion results in a protein that lacks amino acids 219–242 located in the C-terminal extracellular domain. Three amino acids that are critical for susceptibility of *L. lactis* and *L. garvieae* to class II bacteriocins are conserved in the wildtype MptD protein of *L. monocytogenes* EGDe. Two of the three amino acids (Y219 and G242) are located in the domain missing in the MptD protein of the mutant EGY2. We thus made use of the mutant EGY2 to test the antimicrobial compound present in supernatants of *P. acidilactici* M1 and *L. garvieae* 17. *L. monocytogenes* EGDe but not the mutant EGY2 was strongly inhibited by both supernatants and CIEX purified antimicrobial peptides of *P. acidilactici* M1 and *L. garvieae* 17 and the control strain *P. acidilactici* 347 known to produce pediocin PA-1 [56]. Indeed, this suggests that *P. acidilactici* M1 and *L. garvieae* 17 produce class II bacteriocins targeting the PTS^Man^.

## 4. Materials and Methods

### 4.1. Strains and Culture Conditions

Type strains and previously published strains of bacteria and their relevant characteristics used in this study are listed in Table 2. *Listeria sp.* strains were grown on BHI medium at 37 °C. *Pediococcus acidilactici* 347 was grown on MRS medium at 30 °C. Bacteria were routinely cultivated under aerobic conditions on an incubator shaker (I26R, Eppendorf, Hamburg, Germany; 150 rpm). For isolation of LAB, a sample of raw, unpasteurized milk of a cow was diluted and plated on GM17 and MRS agar and incubated at 30 °C. Single colonies were passaged three times on the respective medium to obtain pure cultures. Pure cultures were successfully obtained for nine strains isolated on MRS and 46 strains isolated on GM17 medium (i.e., a total of 55 isolates) (Appendix A).

For growth curve experiments, a 5 mL MRS pre-culture was inoculated with a single colony of respective isolates and incubated at 30 °C and 150 rpm. Following o/N growth, pre-cultures were diluted to an OD_600_ of 0.2 in 100 mL fresh MRS medium in a 250 mL Schott glass bottle and incubated at 30 °C and 100 rpm. OD_600_ and pH were measured at the indicated time-points during cultivation. At t = 3, 12, 24, and 48 h, 2 mL samples of culture broth were harvested and centrifuged at 11,000 rpm× *g* for 5 min to obtain the cell-free culture supernatant for subsequent assays.

### 4.2. Identification of Raw Milk Isolates

For taxonomic identification of the raw milk isolates, strains were grown on agar plates of the medium that was used for isolation and incubated overnight (o/N) at 30 °C. A single colony was resuspended in 1 mL HPLC grade water and samples were analyzed by matrix-assisted laser desorption ionization–time of flight mass spectrometry (MALDI-TOF MS) with a MALDI Biotyper Microflex system (Bruker Daltonics GmbH, Bremen, Germany), a well-established and widespread tool for the rapid identification of microbial isolates in clinical microbiology [58,59] An ethanol/formic acid sample preparation protocol was used as described elsewhere [60]. The spectrum obtained for each strain, representing largely the ribosomal proteins, was compared to three main Biotyper spectra (MSPs) libraries: an MBT compass library (revision F, version 9) containing 8468 MSPs, a filamentous fungi library (revision No. 1) containing 468 MSPs, and a SR library (revision No. 1) containing 104 MSPs, using the software associated with the Biotyper system according to the manufacturer’s instructions. Identification scores of ≥2.3 indicate a reliable identification to species and genus level, score values between 2.0 and 2.3 represent a probable identification to species level, score values between 1.7 and 2.0 represent a reliable genus level, and scores of <1.7 are regarded as unreliable.

For 16S rRNA gene sequencing, genomic DNA was prepared using the GeneEluteTM Bacterial Genomic DNA Kit (Sigma-Aldrich, Taufkirchen, Germany) according to the recommendations of the manufacturer with minor modifications. To increase yield, 5 mL of an o/N culture was used, and DNA was eluted in 50 µL elution buffer. Genomic DNA was used as a template for amplification of 16S rRNA genes using the Q5^®^ High-Fidelity DNA Polymerase, universal primers 27f (AGAGTTTGATCCTGGCTCAG) and 1492r (GGTTACCTTGTTACGACTT) and standard PCR cycling conditions. PCR products were purified using the DNA Purification Kit (Macherey-Nagel, Düren, Germany) and sequenced by Sanger sequencing by a commercial service provider (Microsynth Seqlab GmbH, Göttingen, Germany) using the same primers. The obtained sequences were combined to one sequence for the full length PCR product for each strain, analyzed using NCBI nucleotide BLAST [61], and the sequence hit in the database with the highest identity score was selected for the species identification of the isolate. As the control, *P. acidilactici* 347 was included in both typing approaches.

### 4.3. Screening Procedures

To identify isolates that release antimicrobial compounds, their spent culture supernatants were tested using two different methods in 96-well microtiter plates. To produce supernatants for screening assays, a single colony of each isolate grown o/N on an agar plate was inoculated into 5 mL of MRS medium in a glass tube and cultures were incubated o/N at 30 °C and 150 rpm on an incubator shaker (I26R, Eppendorf, Hamburg, Germany). Aliquots of 100 µL of a preculture were then used to inoculate main cultures of 5 mL MRS, which were incubated in the same way for 16 h. Supernatants were harvested by centrifugation (4200× *g*, 4 °C, 15 min) and stored at –20 °C until further use.

Supernatants were screened using *L. monocytogenes* EGD-e/pNZ-P_help_-pHluorin as an indicator strain. A single colony of the indicator was picked from a freshly grown agar plate, inoculated into 5 mL of BHI containing chloramphenicol (12.5 µg/mL) in a glass tube, and cultivated o/N at 37 °C with aeration (150 rpm).

To screen for growth inhibition, culture supernatants of the isolates or control strains were distributed in 50 µL aliquots into individual wells of 96-well microtiter plates (Standard F; Sarstedt, Nümbrecht, Germany). Then, 150 µL of fresh MRS medium was added (i.e., a 1:4 dilution), mixed, and 100 µL was again removed. As controls, sterile MRS with (positive control) or without (negative control) 0.005% (*w*/*v*) CTAB were included on each plate of the screen. Where indicated, MRS containing pediocin PA-1 (Sigma-Aldrich, Taufkirchen, Germany) at a final concentration of 200 ng/mL was used. The o/N culture of the indicator strain was diluted (1:25) in fresh sterile BHI medium and 100 µL of this suspension was added to each well of the 96-well screening plates. Plates were incubated at 37 °C and OD_600_ was recorded after 7 h.

Screening for membrane damage using the pHluorin assay was performed in a similar fashion as described previously [31]. The assay uses the pH-dependent fluorescent protein pHluorin, which can be used to detect membrane damaging compounds by the change in intracellular pH_i_ [31]. For the screening experiments, 100 µL of supernatants of isolates and control strains as well as control media were distributed into black 96-well microtiter plates (ELISA plate black Med. Bind., F; Sarstedt, Nümbrecht, Germany). Bacteria of an o/N culture of *L. monocytogenes* EGD-e/pNZ-P_help_-pHluorin were pelleted by centrifugation (3000× *g*; 10 min, 4 °C), washed once in phosphate buffered saline, and resuspended in LMB buffer at pH 6.5 [31] to an OD_600_ of 3. A total of 100 µL of this suspension was added to each well of the 96-well screening plates. The microtiter plates were vortexed for 10 s on a Titramax 100 (Heidolph Instruments GmbH & CO. KG, Schwabach, Germany) at 900 rpm, wrapped in aluminum foil, and incubated for 1 h at RT in the dark. Fluorescence was measured with excitation at 400 and 470 nm and emission at 510 nm. The ratio of emission intensities after excitation at 400 and 470 nm were calculated.

### 4.4. Purification and Chromatography of Antimicrobial Compounds

Potential bacteriocins of two isolates were partially purified by ammonium sulfate precipitation and subsequent cation exchange chromatography. For this purpose, 10 mL MRS pre-culture was inoculated with a single colony of respective isolates and incubated at 30 °C and 150 rpm. Following o/N cultivation, the entire pre-culture was used to inoculate 500 mL MRS broth in a 1 l Schott glass bottle. These cultures were incubated for 18 h at 30 °C and 100 rpm. Then, cell-free supernatants were prepared by centrifugation (8000 rpm× *g*, 4 °C, 45 min) and supernatant proteins were precipitated by adding ammonium sulfate gradually at 4 °C with constant stirring to the supernatants until a saturation of 40% (*w*/*v*) ammonium sulfate was reached. The mixture was incubated o/N at 4 °C under constant stirring. Precipitated proteins were harvested by centrifugation (8000 rpm× *g*, 4 °C, 60 min) and the supernatant was discarded. The pellet containing precipitated peptides was dissolved in 50 mL HPLC-grade water and pH was adjusted to 3.9 to match the starting conditions of the cation exchange chromatography.

To further purify potential bacteriocins, cation exchange chromatography using a HiPrep SP FF 16/10 column (Cytiva, Freiburg, Germany; 20 mL volume capacity, 5 mL/min flow rate) was performed. The ÄKTA pure system (Cytiva, Freiburg, Germany) was employed for liquid chromatography. Before applying the sample, the column was equilibrated with five column volumes (CV) of equilibration buffer (20 mM sodium dihydrogen phosphate, pH 3.9). Then, the sample was loaded onto the column and unbound material was removed by washing with 5 CV wash buffer (20 mM phosphate buffer, pH 6.9). Subsequently, the potential bacteriocins were eluted stepwise with 3 CV elution buffer (20 mM phosphate buffer, 2 M sodium chloride, pH 6.9). After each run, the column was extensively cleaned following the instructions of the manufacturer.

### 4.5. Antimicrobial Activity Assays

Bacteriocin activity in different samples was determined as described previously [62]. Briefly, two-fold serial dilutions of the supernatants and other samples were analyzed with the growth inhibition assay described above. One bacteriocin unit (BU) is defined as the reciprocal of the highest dilution showing at least 50% growth inhibition of the indicator strain. BU/mL is then calculated using volumes of the samples in the assay and dilution factors. Where indicated, supernatants were incubated for 30 min at 80 °C or treated with 0.5 mg/mL of proteinase K for 3 h at 37 °C. Proteinase K was inactivated by incubation for 15 min at 80 °C prior to the experiments.

In some experiments, antimicrobial activity was analyzed using a spot-on-lawn assay. For this assay, o/N cultures of sensor bacteria were diluted to an OD_600_ of 0.005 in hand-warm BHI agar. Agar containing sensor bacteria was poured into Petri dishes, and dried for 45 min under a clean bench. Then, 10 µL of a sample (supernatant of producer strains or CEIX purified peptides) was spotted on these agar plates. Following incubation o/N at 37 °C, agar plates were imaged using an iBright^TM^ FL1000 Imaging System (Thermo Fisher, Langenselbold, Germany). Cell-free culture supernatants of producer strains were pasteurized (15 min, 80 °C) prior to the assay to avoid contamination with producer bacteria.

### 4.6. Statistical Analysis

Data of the screening experiments were analyzed by one-way ANOVA with Bonferroni post-test to calculate *p*-values adjusted for multiple. Values obtained with supernatants of the non-bacteriocin producer *L. lactis* MG1363 were set as the control condition. Differences between experimental groups (supernatants) were considered significant at *p*-values < 0. Statistical analysis and visualization of data were performed using GraphPad Prism (version 6.07).

## 5. Conclusions

In summary, our results present a set of tools for the screening of microbial collections for potential producers of antimicrobial activity and subsequent purification and characterization of the antimicrobial compounds. The pHluorin-based assay provides a simple and fast readout for compounds that disrupt membrane integrity, but bacteriocins with bacteriostatic activity are not detected. However, in combination with a second, growth-dependent screening discrimination between strains/compounds with bacteriostatic and bacteriolytic activity is possible. Both screening approaches are limited by the sensor bacteria and their sensitivity to different classes of bacteriocins but can be easily transferred to a wide range of (genetically accessible) microorganisms.

In our study, we focused on a rather small collection of bacteria isolated from raw milk and *L. monocytogenes* as the target/sensor organism but both screening approaches are transferrable to other sensor bacteria and larger collections from other sources (e.g., probiotic candidates in industrial collections). Moreover, we propose also considering classical starter or protective cultures as potential probiotics. Per recommendation of the FAO and WHO of the United Nations, probiotics are defined as “live microbial food supplements which when administered in adequate amounts confer a health benefit on the host” [63]. This includes any bacterium that elicits a positive effect on host health. In sort of a “three birds with one stone” scenario, our set of tools may allow for the identification of bacterial strains that combine characteristics of a starter culture with those of protective cultures and anti-infective probiotics.

## Figures and Tables

**Figure 1 ijms-22-08615-f001:**
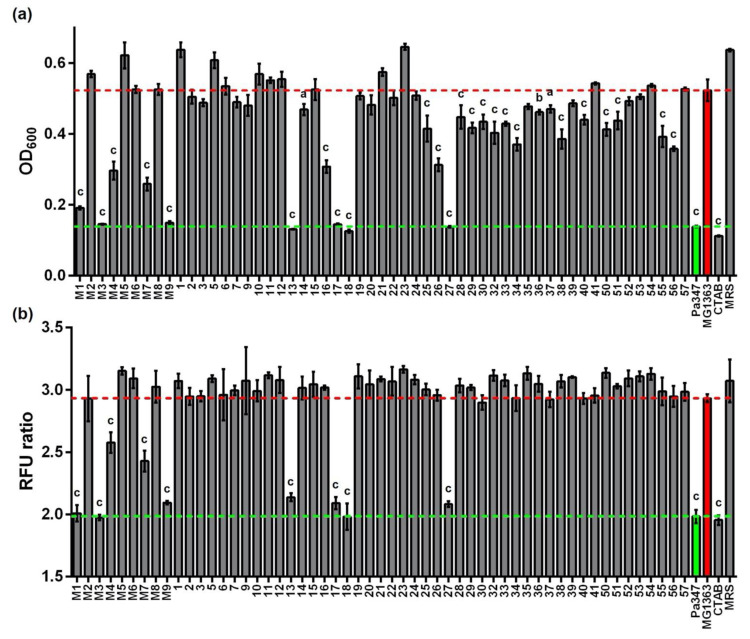
(**a**) Growth inhibitory and (**b**) membrane damaging activity in supernatants of LAB isolated from raw cow’s milk. LAB isolates were grown in 5 mL MRS medium in glass tubes. Activity was measured in culture supernatants after o/N growth using *L. monocytogenes* EGDe/pNZ-P_help_-pHluorin as an indicator. Values are OD_600_ of the indicator strain (**a**) or ratio of fluorescence intensity (RFU ratio; emission at 510 nm, b) after excitation at 400 and 470 nm and are mean ± standard deviation (SD) of *n* = 3 independent experiments. Supernatants of the pediocin producer *P. acidilactici* 347 (Pa347) and the non-bacteriocin producer *L. lactis* MG1363 were used as biological controls. The broken red and green lines indicate OD_600_ of the positive (i.e., complete inhibition of growth by CTAB) or negative (i.e., sterile MRS medium) controls), respectively. Statistical analysis was performed using one-way ANOVA with the Bonferroni post-test to calculate *p*-values adjusted for multiple comparisons and values obtained with supernatants of the non-bacteriocin producer *L. lactis* MG1363 set as the control condition (a: *p* < 0.05; b: *p* < 0.01; c: *p* < 0.001).

**Figure 2 ijms-22-08615-f002:**
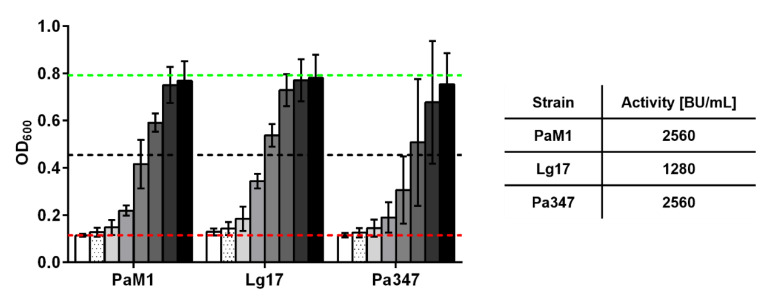
Inhibition of growth of *L. monocytogenes* EGDe/pNZ-P_help_-pHluorin by 2-fold serial dilutions (1:8–1:1024, indicated by a white-black scale) of supernatants of LAB isolates *P. acidilactici* M1 (PaM1), *E. faecium* M4 (EfM4), *L. garvieae* 17 (Lg17), and *L. lactis* 18 (Ll18) or *P. acidilactici* 347 (Pa347). Bacteria were grown o/N in 5 mL MRS medium in glass tubes. Values are OD_600_ of the indicator strain ± SD of *n* = 4 independent experiments. The broken red and green lines indicate OD_600_ of the positive (i.e., complete inhibition of growth) or negative (i.e., sterile MRS medium) controls, respectively. The broken black lines represent growth inhibition of 50% (i.e., the threshold to calculate bacteriocin units).

**Figure 3 ijms-22-08615-f003:**
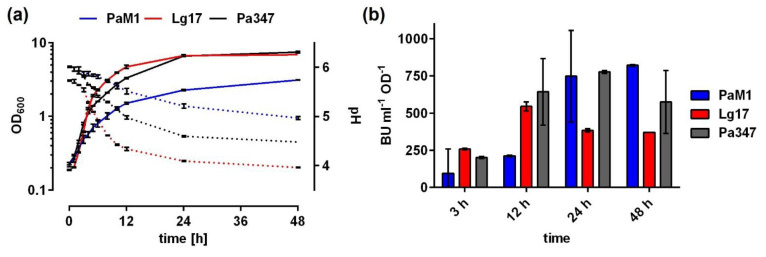
(**a**) Growth (OD_600_; solid lines) and pH (dotted lines) cultures of *P. acidiclactici* M1 (blue), *L. garvieae* 17 (red), or *P. acidilactici* 347 (black) grown in MRS medium. (**b**) Relative antimicrobial activity (BU ml^−1^ OD^−1^) in supernatants of the cultivations shown in (**a**) collected at the indicated time points. BU ml^−1^ were calculated based on the highest dilution showing at least 50% of growth inhibition of the sensor strain (*L. monocytogenes* EGDe/pNZ‒P_help_‒pHluorin) and were normalized to the OD_600_ of the culture at the respective timepoint. All values are mean ± standard deviation of *n* = 3 independent cultivation.

**Figure 4 ijms-22-08615-f004:**
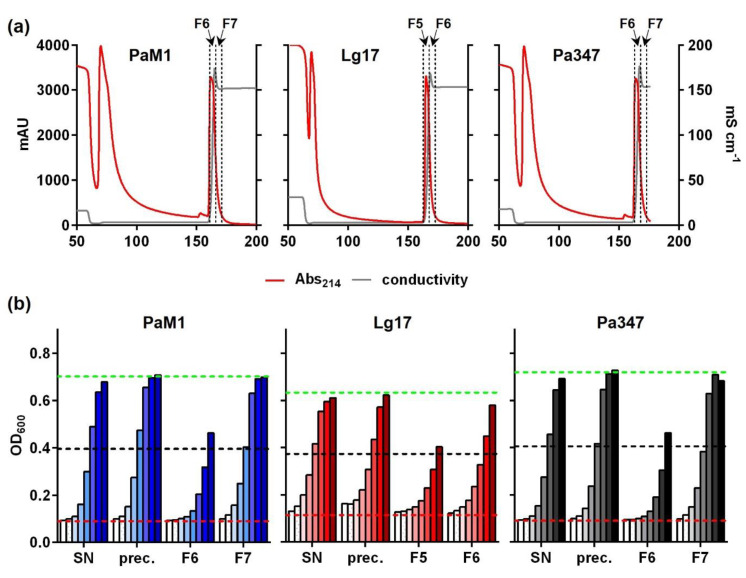
(**a**) Cation exchange chromatography of ammonium sulfate precipitated supernatant proteins of *P. acidilactici* M1 (PaM1), *L. garvieae* 17 (Lg17), and *P. acidilactici* 347 (Pa347) following growth in MRS medium. Red lines indicate absorbance at 214 nm in milli arbitrary units (mAU) and grey lines the conductivity in mS/cm. Broken vertical lines indicate the boundaries of fractions F6 and F7 of the eluate collected for further analysis. (**b**) Inhibition of growth of *L. monocytogenes* EGDe/pNZ-P_help_-pHluorin by 2-fold serial dilutions of different steps of the purification of antimicrobial compounds of *P. acidilactici* M1 (PaM1), *L. garvieae* 17 (Lg17), and *P. acidilactici* 347 (Pa347). SN: spent culture supernatants; prec: ammonium sulfate-precipitated supernatant proteins; F5, F6, F7: fractions 5, 6, or 7 of the elution step of cation exchange chromatography of precipitated supernatant proteins shown in (**a**). Dilutions were 1:16–1:2048 and are indicated by a color scale with increasing intensity. Values are OD_600_ of the indicator strain and are mean ± standard deviation of duplicate measurements of one representative preparation for each strain. The broken red and green lines indicate OD_600_ of the positive (i.e., complete inhibition of growth) or negative (i.e., sterile MRS medium) controls, respectively. The broken black lines represent growth inhibition of 50% (i.e., the threshold to calculate bacteriocin units).

**Figure 5 ijms-22-08615-f005:**
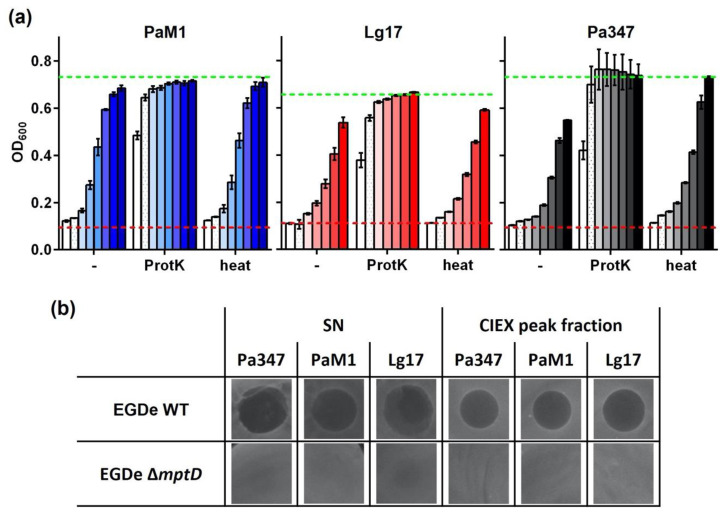
(**a**) Supernatants of *P. acidilactici* M1 (PaM1) and *L. garvieae* 17 (Lg17) or *P. acidilactici* 347 were treated for 3 h with proteinase K (ProtK) or incubated for 30 min at 80 °C (heat) and activity was compared to the untreated control samples (-). Dilutions were 1:8–1:1024 and are indicated by a color scale with increasing intensity. Values are OD_600_ of the indicator strain and are mean ± standard deviation (**b**) of *n* = 3 experiments. The broken red and green lines indicate OD_600_ of the positive (i.e., complete inhibition of growth) or negative (i.e., sterile MRS medium) controls, respectively. (**b**) Spot-on-lawn assays to determine antimicrobial activity in cell-free culture supernatants (SN) or peak fractions collected during CIEX chromatography of ammonium sulfate-precipitated supernatant proteins (shown in Figure 4a) using either *L. monocytogenes* EGDe wildtype (EGDe WT) or its isogenic mutant EGY2 with an 84 bp deletion in the *mptD* gene (EGDe Δ*mptD*).

**Table 1 ijms-22-08615-t001:** Antimicrobial activity in samples of different steps of the purification of antimicrobial compounds of *P. acidilactici* M1 (PaM1), *L. garvieae* 17 (Lg17), or *P. acidilactici* 347 (Pa347). SN: spent culture supernatants; prec.: ammonium sulfate-precipitated supernatant proteins; F6, F7: fractions 6 and 7 of the elution step of cation exchange chromatography of precipitated supernatant proteins shown in Figure 4. Activity was calculated as BU/mL based on the highest dilution showing at least 50% of growth inhibition in the experiment shown in Figure 4. Total activity in each sample (total BU) was calculated using the BU/mL values and the total volume of each sample. Additionally, the recovery in % of the total activity of the entire supernatant of each culture is provided.

Sample	Volume [mL]	PaM1		Lg17		Pa347	
		BU/mL	Total BU (Recovery)	BU/mL	Total BU (Recovery)	BU/mL	Total BU (Recovery)
SN	500	5120	2,560,000 (100%)	2560	1,280,000 (100%)	5120	2,560,000 (100%)
Prec.	50	2560	128,000 (5%)	2560	256,000 (20%)	2560	128,000 (5%)
F6 ^a^	5	20,480	102,400 (4%)	20,480	102,400 (8%)	20,480	102,400 (4%)
F7 ^a^	5	2560	12,800 (0.5%)	2560	51,200 (4%)	5120	12,800 (0.5%)

^a^: For Lg17, CIEX fractions 5 and 6 were used as indicated in Figure 4a.

**Table 2 ijms-22-08615-t002:** Bacterial strains used in this study.

Strain	Relevant Characteristic	Source/ Reference
*Listeria monocytogenes*		
EGDe	Wildtype strain, serotype ^1/2^ a type	[57]
EGY2	EGDe derivative carrying a deletion of 84 bp in the mptD gene	[37]
EGDe pNZ-P_help_-pHluorin	EGDe derivative harboring plasmid pNZ-P^help^-pHluorin for constitutive expression of pHluorin	[31]
*Pediococcus acidilactici* 347	Natural producer of pediocin PA-1; isolated from Spanish dry fermented sausages	[56]

## Data Availability

All data presented in this study are available in the main body or Appendix A of the manuscript.

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
