# Peer review of "Identification of Potential Probiotics Producing Bacteriocins Active against Listeria monocytogenes by a Combination of Screening Tools"

_ijms, 2021, doi:10.3390/ijms22168615_

Round 1
Reviewer 1 Report
The manuscript prepared by Desiderato et al presents a new screening method of microbial collection with potential antimicrobial activity.
The paper is well written and the results well presented.
I only suggest to the authors to add some details about mass spectrometry analysis (e.g. resolution of mass spectrometer, data dipendent analysis, etc..).
Author Response
We appreciate the overall positive perception of the manuscript.
Additional information on MS analyses for identification of bacterial isolates has been provided in the revised manuscript (see lines 367-388 of the revised manuscript)
Reviewer 2 Report
Very interesting and detailed Manuscript. Congratulations to the authors. I have very much enjoyed the work. I have just a few minor comments that may add to the value of te Manuscript:
- Observations as p. 3, row 143, something interesting are more suitable for the discussion section
- In the Methods section please add a description of statistical analysis and data presentation
- Unbold Figures in text
- P. 6, rows 198 etc. are more suitable for the Methods section, also p7, rows 236 etc
- Can the limtations of research be summarised and outlined at the end of discussion? Some are mentioned throught the discussion.
Author Response
We appreciate the overall positive perception of the manuscript. We carefully address all comments raised to the best of our ability as summarized below.
- Observations as p. 3, row 143, something interesting are more suitable for the discussion section
Authors’ response:
We generally agree that interesting observations are usually more suitable for the discussion. In this case, however, we want to make the point that the 9 hits identified by the pHluorin assay are identical to the top 9 hits in the growth dependent assay. The overlap between the hist in both assays is the basis for the selection of strains for further analysis. We thus opted to leave this statement in the results section.
- In the Methods section please add a description of statistical analysis and data presentation
Authors’ response:
A subsection describing statistical analysis and data visualization has been added to the revised manuscript (see lines 480-486 of the revised manuscript).
- Unbold Figures in text
Authors’ response:
References to Figures and Tables are changed to normal font.
- 6, rows 198 etc. are more suitable for the Methods section, also p7, rows 236 etc.
Authors’ response:
As suggested, experimental details in line 198 and 236 were deleted and are provided in the methods section.
- Can the limtations of research be summarised and outlined at the end of discussion? Some are mentioned throught the discussion.
Authors’ response:
Good point. We have slightly modified and amended the conclusions to include a summary of the limitations of the screening methods (see lines 490-496 of the revised manuscript).